# Where Do We Stand on Cervical Spine Immobilisation? A Questionnaire among Prehospital Staff

**DOI:** 10.3390/jcm13082325

**Published:** 2024-04-17

**Authors:** Pascal Gräff, Lisa Bolduan, Christian Macke, Jan-Dierk Clausen, Stephan Sehmisch, Marcel Winkelmann

**Affiliations:** Department of Trauma Surgery, Hannover Medical School, Carl-Neuberg-Straße 1, 30625 Hannover, Germanywinkelmann.marcel@mh-hannover.de (M.W.)

**Keywords:** cervical spine immobilisation, cervical collar, prehospital, multiple trauma

## Abstract

**Background**: Cervical collars (CC) are routinely used in prehospital trauma treatment. However, over the past years, their application was discussed more critically since they increase intravenous pressure due to reduced venous drainage and the possibility of secondary cervical spine injury. Guidelines have been adjusted accordingly. The question is how efficient has this been put into practice, and how good, as well as up to date, is the knowledge of prehospital emergency medicine personnel about indications on cervical spine immobilisation? **Methods**: A 15-item questionnaire regarding the self-evaluation and result checking of the right indications for the use of a cervical collar in the prehospital setting was sent to paramedics and emergency doctors in Germany. Two hundred and nineteen completed surveys were statistically analysed. **Results**: Mean age of the participants was 30.45 ± 8.8. 72% were male. Regarding subjective safety, the appropriate indication of CC participants reached 79.8 ± 19.5 on a metric scale from 0 (no safety) to 100 (full safety). Mean right answers were as follows: Ambulance man (RS) 0.78 ± 0.84, paramedic (RA) 0.9 ± 0.74, paramedic (NFS) 1.03 ± 0.83 and emergency doctor (ED) 1.75 ± 1.06 (*p* = 0.013, Kruskal–Wallis Test). Participants who estimated their knowledge < 85% had 0.83 ± 0.8 right answers, and > 85% had 1.14 ± 0.9 right answers. **Conclusions**: Rational spine immobilisation is still necessary in severely injured patients. This study highlights the importance of continuing education using ongoing training, lectures or online learning with a questionnaire as a monitor for success to ensure the transfer of evidence-based medicine into daily practice.

## 1. Introduction

Immobilisation of the cervical spine has been a topic of high importance for a long time. Since the beginning of preclinical trauma treatment, the immobilisation of the cervical spine has been a high priority. Due to the importance of this matter, many different tools have been introduced to achieve this goal, many of which are still in use today. The first ones to be invented were cervical collars (CC) that aim at the total immobilisation of the cervical spine and are used throughout various countries even today. Initially, it was invented almost 80 years ago and has had huge success since it was promoted by the American College of Surgeons (ACS) and the Advanced Trauma Life Support (ATLS) program [1]. In the beginning, the use of a rigid CC was performed almost without limitations. Over the past 10 years, the application of CC to every trauma patient has been critically judged due to higher intracranial pressure [2] because of impaired venous drainage [3]. This fact was described in detail in 1991 when Craig and Nielsen published their study reporting this matter [2]. However, the use of rigid CCs in a prehospital and early intrahospital trauma situation has continued until today without any relevant decrease.

Moreover, increased intracerebral pressure is not the only problem with CC. The movements of an unstable spine are not properly reduced at all [4]. As a matter of fact, the movement of the upper cervical spine can be further increased [5]. Although CCs have several limitations, the alternatives are limited.

Rigid CCs are still in use, although one can never know whether they are effective in reducing secondary spinal cord injuries and how many of them might have been prevented [6].

Even though some regions have reduced or stopped recommending routine use of these collars, emergency medicine providers still immobilise the cervical spine with collars, frequently assuming that CC reduces cervical spine motion even better compared to no immobilisation [7]. Also, the practicability, easy storage, quick-to-perform application, and flexibility are unmatched. The next step to reduce inappropriate usage is to improve algorithms on CC applications.

To sum up, we seem to have a good understanding of the benefits and problems of CC. In addition, there are alternative ways of immobilisation that have been used over the last decade. What follows is a brief examination of them.

This analysis begins with the Kendrick Extrication Device (KED), also called a “rescue corset”. By popular opinion, it is believed that it has the capacity to immobilise the cervical spine. Using a search in established medical databases, only one article deals with the immobilisation of the cervical spine, in which it is especially mentioned as an additional tool without the ability to immobilise independently [8]. Furthermore, it is, as the manufacturer describes the correct usage, only a tool for the rescue or, to be more precise, the extrication of patients. By looking at the homepage of Ferno^®^ (Wilmington, NC, USA), the manufacturer of the KED, the additional use of a cervical collar is proposed, which is backed by publications, which support and promote this way of usage [9]. One article by Engsberg et al. examining the efficiency and method of cervical spine immobilisation during an extrication supports the combination of a KED with a Cervical Collar [10]. An immobilisation of the cervical spine without a CC is not mentioned. This supports the assumption that the standalone KED is not an appropriate tool for the above-mentioned purpose.

Another opportunity for spine immobilisation is the vacuum mattress. Initially designed for exactly this purpose, a couple of studies especially described the benefit of immobilisation using a mattress compared to no immobilisation [11]. The cervical immobilisation can be achieved with and without using headblocks with supposedly different efficiency, which has not been scientifically examined. However, it is used as a form of good clinical practice in some regions.

The third way to immobilise the spine and, in particular, the cervical spine, which is commonly used at least in Germany, is a spine board. This is a board made from hard plastic on which patients are strapped down in order to immobilise them. The cervical spine is routinely immobilised using head blocks that can be attached to the board. The efficiency of cervical spine immobilisation is examined in a few studies with varying results. Nolte et al. preferred the spine board for the purpose of immobilisation [12], and Prasarn et al. [13] favoured the vacuum mattress since it provided a better immobilisation of the cervical spine. It should be noted that Nolte et al. used volunteers, whereas Prasarn et al. [13] used cadavers. Moreover, Nolte et al. used a CC in combination with a vacuum mattress. An undesirable side effect of using spine boards is the risk of pressure ulcerations. Oomens et al. and Sheerin et al. described that pressure ulcerations are to be found in spine boards, with the possibility of reducing them using a softer comfort board [14,15]. Nevertheless, after this fact was first described, the spine board was rapidly taken out of consideration for the strapped-down transport of injured patients. As an alternative the vacuum mattress was restored as the primary measure for immobilising patients, commonly used with a CC. As the CC still has the above-described problems of increased intracerebral pressure the next idea was the combination of the vacuum mattress with headblocks.

Although this description is of little relevance, it shows how quick and sometimes unverified changes are put into action. Prehospital medicine is far more shaped by experience than other medical branches. Often, procedures and practices are introduced at first and verified afterwards.

Therefore, it seems reasonable to clearly define the appropriate indications and application of cervical spine immobilisation with CC and other measures of immobilisation in a prehospital setting. Many countries, as well as Germany, manage the indications and usage by guidelines. Experience has shown that the perceived use in daily routine differs vastly from the recommended. A rationale could be inappropriate knowledge and protracted implementation of guidelines.

As a decision-making tool on whether cervical immobilisation should be applied or not, the Nexus criteria or the Canadian c-spine rule are frequently used. Originally designed to have a tool for clinically examining the necessity for an X-ray of the cervical spine [16], these criteria can be used to decide about the need to immobilise the cervical spine. This arises from the simple assumption that a cervical spine does not need immobilisation if relevant injuries can be ruled out without the necessity of medical imaging.

The purpose of this study is to find out how well changes in recommendation of cervical spine immobilisation are implemented and spread in the community of prehospital emergency medicine providers. On the one hand, our main interest focuses on the actual appropriate application of spine immobilisation, and on the other hand, we are convinced that this study can be a surrogate measure of the latency between implementation and actual pervasion of research findings in daily routine.

## 2. Materials and Methods

To answer this question a survey was created consisting of 15 questions. German paramedics of different levels of training and education, as well as prehospitally active emergency doctors, were requested to participate in that survey. The paramedic system in Germany consists of different educational levels. There are “Rettungssanitäter” (RS; ambulance man/officer) who have to undergo a three-month training. The vast majority of paramedics are “Notfallsanitäter” (NFS; paramedic), which is the successive version of a “Rettungsassistent” (RA; paramedic). Both had to complete training for 2–3 years, followed by a state examination at the end. The RA is the older version, with less competence and authorisation to, for example, administer drugs. For NFS, there was a long-lasting programme allowing the RA to upgrade their status to NFS with only a couple of weeks of extra lessons and an examination at the end.

In order to become a prehospital emergency doctor (ED) in Germany, one needs to have at least 2 years of clinical practice and have to complete 80 h of seminars with additional practical experience and a final medical council exam.

This survey was performed with SoSci Survey, a website hosted in Munich, Germany for the main purpose of scientific surveys. The benefit of this website is the fact that it works according to German and European data security regulations, which reduces administrative expenses.

All in all, this questionnaire was divided into three parts. The first part had the goal of obtaining information about our participants, e.g., if they consider themselves to be a specific gender and asking for their age. After that, the questionnaire asked about their medical education, the time they have been working in the emergency service as well as the time since the participants started working in this business. The last question of this part asked about the percentage of occupation in this field, e.g., main occupation or part-time.

The second part regarded the self-evaluation of the participants. This part consisted of five questions with a multiple choice single best answer method to test the knowledge about the immobilisation tools, one sorting question to look for the participant’s assessment of the different methods in immobilisation regarding their effectiveness, one question with 5 sub-questions on the importance of cervical immobilisation via a scaling design, as well as questions regarding the personal details of the participants.

The third and final part was designed to test the participants’ knowledge of indications for cervical spine immobilisation. In this section the question described a scenario with an accident. The answer set always consisted of 5 identical answers in a multiple-choice character. In order to find the correct answer, the nexus criteria were used as well as the aim to exchange the CC as early as possible against the vacuum mattress with head blocks. The use of a spine board was strictly limited to the period of time between the rescue from an inaccessible place until the beginning of the transport in order to reduce the risk of ulcerations. If these three key factors were met the answers could have been answered correctly.

After being online for three months to allow the target group to answer, the survey was closed, and the data were downloaded for analysis as an Excel sheet. The data was stored on the server of the website for three more months and automatically erased. After the download, the data were stored on an external hard drive in a private and inaccessible place for the public.

The statistic was performed using SPSS 27 (IBM, Armonk, NY, USA), and the graphs were created with SigmaPlot 13.0 (Inpixon, Palo Alto, CA, USA). The main test performed was an Analysis of Variance (ANOVA) with a Kruskal–Wallis Test or a T-test combined with a Mann–Whitney Test for the comparison of two groups.

## 3. Results

The total amount of commenced questionnaires was 254, of which 35 had to be sorted out due to a significant number of unanswered questions, leaving 219 for analysis. The mean age of all participants was 30.45 ± 8.8 years, and 72% were male. Regarding subjective safety about the appropriate indication of CC, participants’ self-assessment reached 79.8 ± 19.5 on a metric scale from 0 (no safety) to 100 (full safety).

The professions were almost even distributed between RS and NFS, whereas RA and ED both had a much smaller number of participants (Figure 1).

This graph shows the professions of the participants finishing the questionnaire in an increasing order regarding their educational level. Abbreviations used: RS = “Rettungssanitäter”, RA = “Rettungsassistent”, NFS = “Notfallsanitäter”, ED = Prehospital Emergency Doctor.

Regarding the right indications for CC, there were significant differences between the professional groups (Figure 2) and in correlation with the subjective knowledge of the indications (Figure 3). The mean right answers were as follows: RS (0.78 ± 0.84), RA (0.9 ± 0.74), NFS (1.03 ± 0.83) and ED (1.75 ± 1.06) (*p* = 0.013, Kruskal–Wallis test).

Participants with higher self-evaluation had significantly more mean right answers in the third part of the questionnaire. The cut-off was set to 85 after the dichotomisation of self-evaluation according to the median value. Those participants who estimated their knowledge < 85 had 0.83 ± 0.8 right answers, and >85 had 1.14 ± 0.9 right answers (*p* = 0.027; Mann–Whitney test). There were no differences with regard to the post-qualification experience (*p* = 0.274, Kruskal–Wallis test).

This graph shows the number of right answers depending on the participants’ profession with increasing educational levels. Abbreviations used: RS = “Rettungssanitäter”, RA = “Rettungsassistent”, NFS = “Notfallsanitäter”, ED = Prehospital Emergency Doctor. The differences between the single professional groups were evaluated using pairwise comparisons of the Kruskal–Wallis test and are depicted in the table below the graph.

This graph describes the mean number of right answers with regard to the participants’ self-evaluation. Self-evaluation < 85: 0.83 ± 0.8 right answers compared >85: 1.14 ± 0.9 right answers [*p* = 0.027 (Mann–Whitney test)].

The ranking of different prehospital immobilisation devices according to their expected effectiveness is given in Figure 4. KED was rated 4.1 ± 1.1 on a 5-level scale (1 least effective–5 most effective) to immobilise the cervical spine. CC was rated 3.9 ± 0.9, vacuum mattress 2.9 ± 1.2, spine board with headblocks 2.4 ± 1.1, and scoop stretcher 1.6 ± 1.0, respectively (*p* < 0.001; Friedman test).

One question asked the participants to rank the devices’ ability to immobilise the cervical spine (*p* < 0.001, Friedman test).

## 4. Discussion

This study demonstrates the low implementation and pervasion of the current recommendation of cervical spine immobilisation in trauma in daily routine. However, it points to the dependency of appropriate application on educational level and knowledge-based self-assurance.

Looking at these results, the allocation of gender and age matches the current distribution in German prehospital emergency services, excluding structural and regional differences. The distribution of educational level corresponds with the actual proportion of paramedics and emergency physicians in Germany. In larger cities in Germany, the ratio between paramedics and EDs is around 9 to 1 [17]. Therefore, the likelihood that a paramedic performs the initial prehospital treatment is quite high. This highlights the importance of the correlation between educational level and indicational safety. Although the German paramedic system cannot be transferred to other countries without limitations, it seems reasonable that better education leads to improved guideline-oriented treatment. Despite the fact that a correct application is important to reduce the risk of secondary cervical spine injury, an incorrectly applied collar might worsen the situation [18,19]. Even if the CC is re-applied correctly on the second try, harm might have already been caused because of the varying quality of application [20], leading to a higher rate of complications, which might be prevented by the restricted usage according to the appropriate indication. In our study, even the best participants scored only half of the answers correctly. With decreasing education, the answers were even less likely to be correct. Thus, there seems to be a mismatch between evidence/guidelines and daily routine. This study was not aimed to prove disadvantages referable to inappropriate implementation of guidelines. Therefore, we are not able to assess whether this deviation from guidelines leads to worse treatment or outcomes. Nevertheless, we are convinced that guideline-oriented treatment is beneficial for patients. Therefore, it is essential to increase the degree of awareness. Continuing education is one opportunity, as well as cost, time-effectiveness, and benefit to patients [21]. We cannot make any suggestions on how proper education should be conducted or which mode should be used [22,23]. However, regular obligatory refreshing seems to be essential. In this study, the subjective safety was evaluation almost 80 on a scale from 0 to 100, while the total number of correct answers was quite low. In this study, participants who evaluated themselves as more confident with the indications and usage of CC earned higher scores. By self-assessing with a lower score the participants might have a lower self-esteem or are able to reflect their abilities even better. Knowing that they have less routine in the indications for cervical spine immobilisation and the correct usage of the immobilisation methods could lead to an increased willingness to improve their knowledge about this matter. Therefore, nuanced self-awareness might be a potential leverage point for education. Yet again, we are absolutely convinced that ongoing education and lifelong learning are essential to promote the best patient care. The number of correct answers correlated with educational level accordingly. Regarding the results of the NFS, the gap in the results compared to the RA is not as large as it might be since all of the before mentioned participants, on average, scored around one correct answer, whereas the ED almost scored two and thus almost double the correct answers. The findings of the analysis shown in Figure 4 were unexpected, according to our experience. With the highest number of participants voting for the KED, without a CC, as the best way to immobilise the cervical spine indicates a relevant unconsciousness in this field. As the KED should only be used in combination with a CC, the independent ability to immobilise the cervical spine is minimal. Since the scenario uses only the KED, the patient is provided with a strap on the forehead to secure the head and, thus, prevent the cervical spine from moving. Looking at the further results, this study indicates that there is a relevant lack of knowledge regarding appropriate cervical spine immobilisation. The scoop stretcher-without any further measures for immobilisation, like head blocks, is not unanimously voted in last place but has been voted in third place by some participants.

This left room for much speculation. Our data can only show that there is much backlog but can give no plausible reasoning. The actual training proportion of cervical spine immobilisation is comparable over all different educational levels. Moreover, we did not find any dependency on post-qualification experience. In our opinion, continuing education is of particular importance. However, we are not able to make a point about whether intrinsic or extrinsic reasons are crucial for the degree of success of implementation of treatment recommendations. Intrinsic factors like a lack of willingness to improve or renew one’s knowledge are hard to assess and even more hard to affect. Certainly, no one will be very honest when being asked about personnel’s unwillingness regarding education. We can only speculate about EDs that keep up to date more efficiently compared to paramedics.

Thus, focusing on extrinsic factors might be more beneficial. Despite growing scientific evidence and guidelines, the actual transition into daily work is challenging. High workload, rapid turnover, and a variety of competing training facilities may hamper educational success in a very specific scope [24]. Nevertheless, any effort to implement guidelines and evidence-based medicine is better than no effort [22]. As described in the introduction, one problem is the fast-changing and untested recommendations for the usage of different immobilisation tools. It could be argued that these recommendations stem from “clinical judgment” [25]. The tools for teaching these changes are relatively few. On the one hand, they are distributed via journals or by mouth-to-mouth propaganda. However, Rees et al. [26] were able to demonstrate the benefit of this peer-to-peer teaching in evidence-based medicine; our findings suggest that it might be of little effectiveness in a specific scope. With this realisation, it is worth noting that even though it is taught, it may not necessarily be transferred to clinical practice [27]. This should help the medical providers to understand evidence-based medicine [28], but as this knowledge transfer in German prehospital medicine is hugely driven by personal engagement and interest in the subject, it is impossible to say if the rather bad results in the questionnaire stem from little knowledge or the refusal of applying it to the professional work.

The immobilisation of the cervical spine is currently most likely indicated by the nexus criteria and the Canadian c-spine rule [29]. As it is currently under revision there is no final proposition for the German area. Nevertheless, the use of cervical collars is reduced in favour of the use of a vacuum mattress, which might be added by headblocks. In other recommendations, the use of a spine board is still favoured [30]. The problem with all these guidelines is the fact that there is little evidence, and it is unclear how effective these methods are in the daily routine. Additionally, we are blind at the age boundaries to a certain extent. There are studies reporting the risk for geriatric patients to be undertreated and fractures overseen, for that matter [31]. On the other end of the age spectrum, there are children with insufficient data regarding the accuracy of fracture detection [32].

Karason et al. found out that a Stifneck, which is a product from Laerdal commonly used in Germany, is more effective than a Philadephia collar (most comparable to a California collar) [33]. The efficacy of immobilisation with a collar and a KED might be comparable to the single use of a CC. A beneficial effect of the combination of both has not been verified yet [10]. The KED was deemed to be the most effective in cervical immobilisation by the participants in this study. However, it takes a long time to apply [34]. Furthermore, in paediatric patients, there appears to be no difference between KED and CC [8], whereas in adult patients, a California collar seems to be less effective compared to KED. As mentioned before, the spine board and the vacuum mattress are comparable regarding immobilisation, but a study comparing all of these immobilisation tools has never yet been performed. To clarify the full potential of the various tools, regardless of all the side effects, a comparative study should be conducted with the aim of clarifying the individual potential for cervical spine immobilisation. By knowing the most effective form of immobilisation, the guidelines on the application could be more easily updated and thus give the users a higher degree of safety in the selection of the right tool. Until then, they need to be taught every aspect there is to know regarding the pros and cons of using a CC, spine board or vacuum mattress. Hence, this study brings up another question: How guideline-oriented is our daily medical treatment apart from studies? The pure fact that we have guidelines does not necessarily imply that these guidelines are already well-implemented. In our opinion, this study can be interpreted as a surrogate parameter of the unrecognised or imponderable discrepancy between recommended and actual treatment. We could prove this for the indication-appropriate use of cervical collars only. However, we are fully convinced that this assumption is transferrable to different situations to a variable extent.

## 5. Conclusions

This study demonstrates that there is a substantial lack of knowledge regarding the appropriate application of cervical collars and cervical spine immobilisation in trauma patients. With higher educational levels as well as higher self-evaluation, the knowledge is significantly better. This highlights the importance of continuing education for the transfer of evidence-based medicine into daily practice, for the prehospital emergency staff as well as the physicians working as part of the team.

An essential precondition for adequate teaching is a clear and reliable guideline. For this purpose, further studies comparing common immobilisation tools on the market are necessary in order to provide more safety for the patients and easier processes for the users.

## Figures and Tables

**Figure 1 jcm-13-02325-f001:**
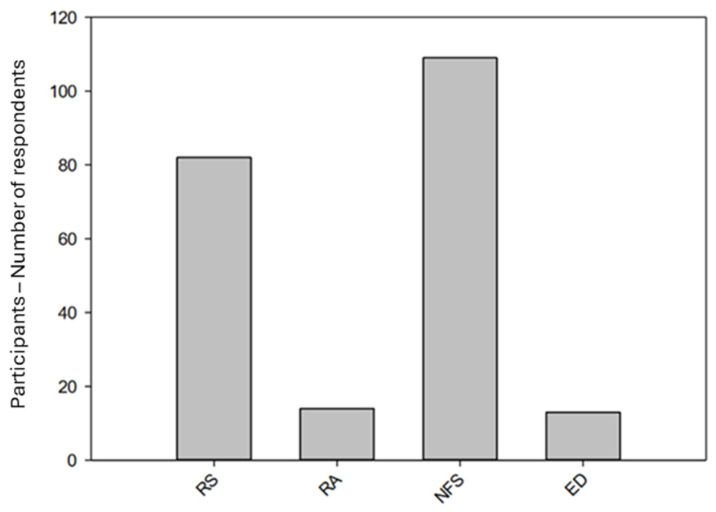
Distribution of participants over the different professions.

**Figure 2 jcm-13-02325-f002:**
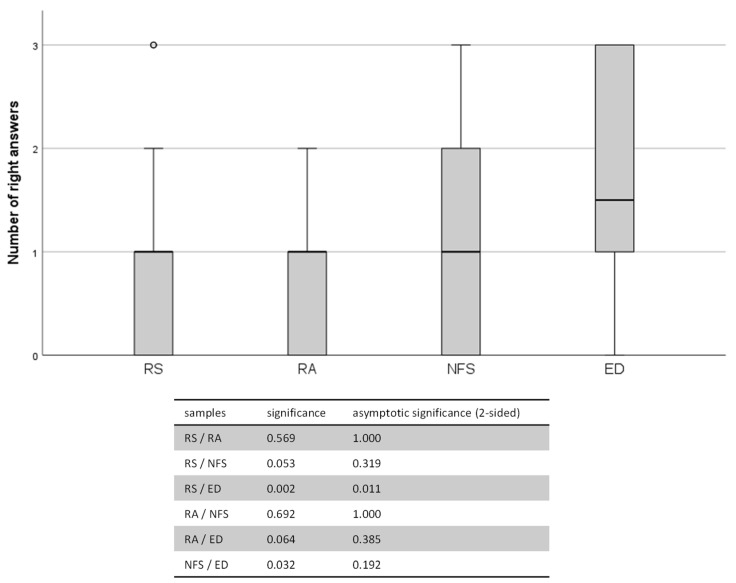
Results of the right answers on control questions.

**Figure 3 jcm-13-02325-f003:**
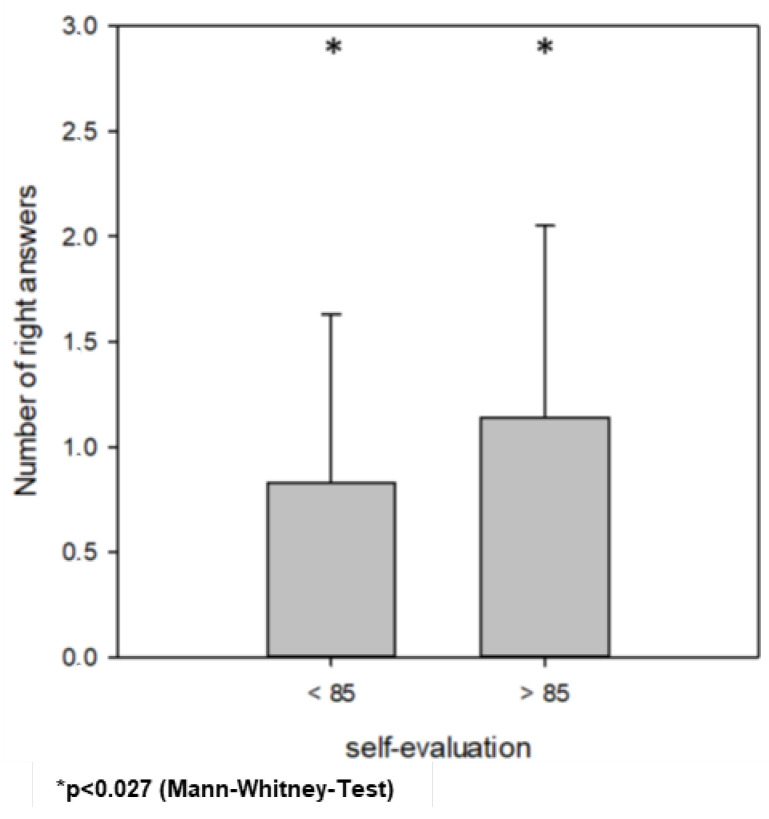
Results of right answers regarding self-evaluation.

**Figure 4 jcm-13-02325-f004:**
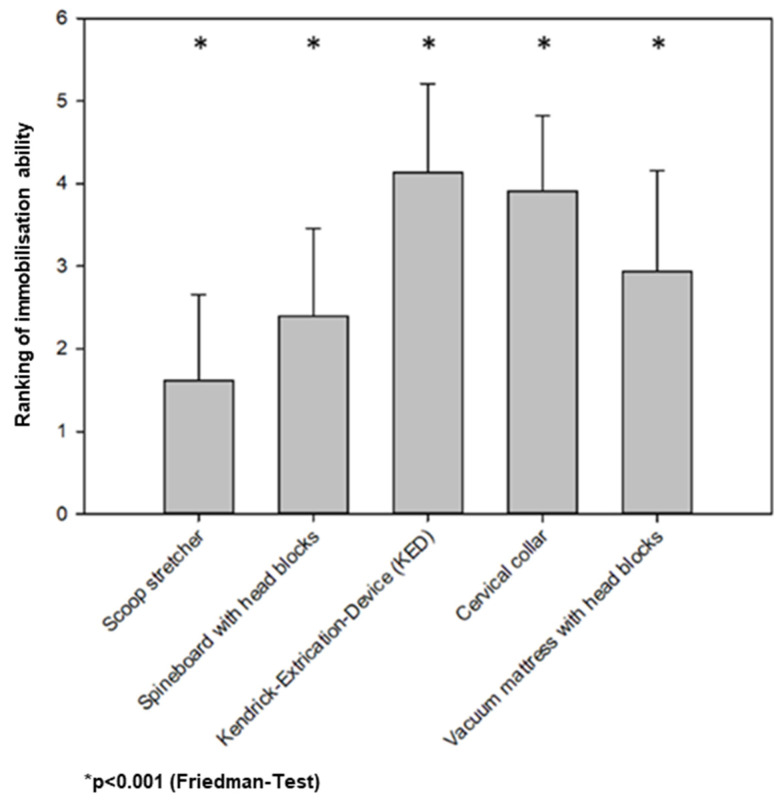
Which of the immobilisation measurements are deemed most effective?

## Data Availability

The datasets used and/or analysed during the current study are available from the corresponding author on reasonable request and are not accessible to unauthorized persons.

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
