# Peer review of "Where Do We Stand on Cervical Spine Immobilisation? A Questionnaire among Prehospital Staff"

_jcm, 2024, doi:10.3390/jcm13082325_

Round 1
Reviewer 1 Report
Comments and Suggestions for Authors
Dear Editors,
in my opinion, the content of the manuscript requires corrections. The quality of this article could be improved with the suggestions below.
INTRODUCTION:
This chapter is an introduction to the purpose of the study, but some facts are mentioned several times. It is worth shortening this chapter.
RESULTS:
Figure 1. - I recommend modifying the chart: removing the duplicate description above the chart; adding on the Y axis next to "participants" - "number of respondents"; adding a legend under the chart (abbreviations used in the chart should be explained so that the chart is readable in itself). Similarly for Figs. 2 and 3.
In the case of Fig. 2 (as well as in the text: page 5, lines: 184 - 187), it should be noted exactly which groups were statistically significant (and what was the level of significance).
Fig. 4 - similarly duplicated description and no "y" axis signature.
The "results" section should present detailed results of the statistical analysis, therefore "evaluative" terms should be avoided and only the statistical (mathematical) basis of the differences should be presented (it is worth paying attention to lines: 205-211).
DISCUSSION:
Speculations and assumptions that are not supported by the available literature should be removed from this section. It is worth supplementing the chapter with a comparison of your own results with the authors of similar scientific publications dealing with the differences in evidence-based medicine and the everyday practice of doctors.
CONCLUSIONS:
Only conclusions drawn from the authors' own research should be included here.
Author Response
Dear Reviewer,
thank you for your advices on how to improve our manuscript. I hope we have changed it to your satisfaction.
I would like to discuss your points:
INTRODUCTION:
We have sharpened this chapter a bit, but it in our opinion it is worth describing the problems which arise with the mentioned progress in this field. By leaving out more of these details the less specialized readers might not understand the problems the paramedics have to deal with an which we were keen on testing
RESULTS:
We hope to have changed to figures the way you meant us to do. Especially with the changes in Fig. 2 the manuscript has improved.
DISCUSSION:
We absolutely agree, that speculations and assumptions should have no part in this manuscript. Nevertheless we wanted to keep a few as they are important to show the reader how difficult navigating in this field is and ongoing, unproven changes are. Furthermore we tried to provide it with an easier to read "story line".
We have added a short section describing the problems with EBM and the everyday practice.
CONCLUSION:
We have worked on the conclusions and deleted the parts not reflecting our own research.
Kind regards
Reviewer 2 Report
Comments and Suggestions for Authors
The article is about an interesting subject. Authors tested knowledge of correct practice regarding the use of cervical immobilization among pre-hospital staff.
The survey may be a useful tool. The Authors highlighted the very low rate of correct answers among participants. This raises some doubt about the clarity of questions; another possibility is that the questions were too difficult or too far from the daily activity of participants.
A specific national guideline is still lacking in this field, but recommendations from international associations are available (see, for example, Zileli M, Osorio-Fonseca E, Konovalov N, Cardenas-Jalabe C, Kaprovoy S, Mlyavykh S, Pogosyan A. Early Management of Cervical Spine Trauma: WFNS Spine Committee Recommendations. Neurospine. 2020 Dec;17(4):710-722. doi: 10.14245/ns.2040282.141. Epub 2020 Dec 31. PMID: 33401852; PMCID: PMC7788428.).
Specific organisation of German emergency system limits generalizability of conclusions.
Author Response
Dear Reviewer,
Thank you for your kind words. We have looked at the recommended example and interestingly it proved our point, that it is difficult to give good advice since it differs from so many other recommendations. It is after all an interesting and quickly changing field.
As to our questions. We do not think, that they are too far from the daily activity or to difficult. We have tested these questions in advance to our study with a small group of doctors from our clinic, we knew they would not take the survey and they had a significantly (although not mathematically tested) higher number of right answers.
We hope this lowers your concern with our questions.
Kind regards
Round 2
Reviewer 1 Report
Comments and Suggestions for Authors
The authors changed the content of the article in accordance with my suggestions. I recommend that the manuscript be accepted for publication.